# Studies on Hot-Rolling Bonding of the Al-Cu Bimetallic Composite

**DOI:** 10.3390/ma15248807

**Published:** 2022-12-09

**Authors:** Ioana-Monica Sas-Boca, Dana-Adriana Iluțiu-Varvara, Marius Tintelecan, Claudiu Aciu, Dan Ioan Frunzӑ, Florin Popa

**Affiliations:** 1Faculty of Materials and Environmental Engineering, Technical University of Cluj-Napoca, 28 Memorandumului Street, 400114 Cluj-Napoca, Romania; 2Faculty of Buildings Services Engineering, Technical University of Cluj-Napoca, 28 Memorandumului Street, 400114 Cluj-Napoca, Romania; 3Faculty of Civil Engineering, Technical University of Cluj-Napoca, 28 Memorandumului Street, 400114 Cluj-Napoca, Romania

**Keywords:** hot rolling of multilayers, Al-Cu composite, diffusion at the interface, severe deformation, bimetallic composite

## Abstract

Through the approaches in this article, an attempt was made to analyze the bonding of Al-Cu bimetallic composite layers and the highlight of the diffusion at the boundary between the layers, by hot rolling. An aluminum alloy 6060 plate (EN-AW AlMgSi) and a Cu-ETP ½ hard (CW004A) plate were used. All of these layers of materials were TIG-welded, at both ends, into a heat-treated layered composite and subsequently subjected to the hot-rolling process. The Al-Cu composite material obtained was analyzed by scanning electronic microscopy (SEM) analysis, after being subjected to the tensile test, as well as energy-dispersive X-ray (EDX) analysis. The obtained results highlighted the diffusion at the boundary between the layers of the Al-Cu composite as well as its ductile breakage and the distribution of the amount of Al and Cu at the interface of the layers.

## 1. Introduction

The discovery of new materials with special properties, against the background of resource depletion and the need to recycle existing materials, always arouses the interest of researchers [1,2,3,4,5,6,7,8,9,10,11,12,13,14] around the world. The production of light-resistant materials, which is necessary for the aeronautical, naval, energy, electrical, and electronic industries, pushes research towards the association of materials with completely different properties [15,16,17,18,19,20,21,22]. Aluminum, copper, and their alloys were used as a matrix in the component of rolling composite materials [23,24,25].

Copper’s properties such as its high thermal and electrical conductivity, high modulus of elasticity, and good strength makes it widely useable for gas and water pipe devices, which allows heat transfer such as boilers, electronic circuits, and electrical components within the fields of industry: energy, aeronautical, naval, and automotive. Copper was used as a reinforcing element in the production of rolling composite materials [8,9,11,17,21,23,24,25], through the ARB process (accumulative roll bonding) [7,9,10,11,12,19,20,21,22,23,24,25] or hot rolling [1,2,17,18]. Along with other materials such as Ti, Sn, Mg, Fe, Nb, and Al, the properties of rolled composites have increased [3,9,15,17,21,23,24,25]. The problem with the bonding mechanism within layers of composite materials was the main subject of numerous studies [1,2,7,9,10,11,12,15,16,17,18,19,20,21,22,23,24,25], throughout the last decades. Regarding the mechanism of the bonding of the layers, the theories were dependent on the choice of the deformation process [3,16,17,18] (in this case rolling), which can take place through hot or cold conditions. These theories have been narrowed down to four: two are attributed to cold bonding [7,10,11,12,19,20,21,22,23,24,25], they include the film theory and the energy barrier theory; and two are attributed to hot bonding [1,3,17,18], these include the theory of diffusion bonding and the theory of diffusion bonding and the joint recrystallization theory.

Research in the field of layered composites has intensified since 2013. Lee K.S. et al. analyzed the influence of the reduction ratio [26]; Kim I.-K. et al. analyzed the effect of component layer [27]; Kocich R. et al. analyzed the fabrication of cold-swaged multilayered [28]; Li X-B. et al. and Yu C. analyzed the roll bonding [29,30]; Jandaghi M.R. analyzed the impact on the corrosion [31]; Ebrahimi M. analyzed the characteristic investigation of the tri-layered composite [32]; and Mo T. et al. [33] and Zhang J.-P. et al. [34] analyzed the interface effect on the microstructure. The microstructure problems were also investigated by Liu T. [35], Shayanpoor A.A. [36], Venugopal S. [37], and Zhu Y. [38]. Cheng, J. et al. proposed a post-weld composite-treatment process for improving the mechanical properties of AA 6061-T6 aluminum alloy welded joints [39].

The main objective of this work is to obtain a multilayer Al-Cu composite by hot rolling and determine the characteristics of this layered composite. This composite could lead to the accumulation of advantages offered by both materials with very different properties and a reduction in their major disadvantages. The obtained material should have well-connected layers, and the entire composite system should be highly resistant to tensile strength and to wear value and consist of ductile properties.

## 2. Materials and Methods

### 2.1. Sample Preparation

The purpose of this work was to create multilayer type samples of copper and aluminum. To make these samples, were used commercial extrusion aluminum from the 6000 series (EN-AW-6060 T6) and Cu-ETP (electrolytic tough pitch copper) ½ hard (CW004A). The multilayering of the composite was made after hot rolling. The samples were made of four layers (Figure 1), two of aluminum and two of copper, which were clamped alternately by front faces welding (Figure 2) type TIG (tungsten inert gas welding) with the addition of aluminum at both ends of the sample to reduce the possibility of sliding between the layers at the entrance to the rolling mill. Before TIG welding, the four layers were mechanically cleaned, degreased, and individually sprayed with a solution of 30% Al_2_O_3_ + 70% H_2_O_2_.

The materials used in this work were chosen for reasons of hardness and resistance to shocks, but at the same time to have a high ductility. The materials used were aluminum EN AW-6060 T6 (AlMgSi) and Cu-ETP ½ hard (CW004A) with chemical composition, presented in Table 1 and Table 2, respectively, and the mechanical properties in Table 3 and Table 4.

The chosen materials were commercial materials delivered in the form of a plate with dimensions of 20 × 3 × 4000 mm.

In the first stage, several pieces of plates of decreasing size were cut from the following materials: aluminum, one plate of 300 mm and one of 290 mm, after cutting two plates of copper, one of 295 mm and one of 285 mm, which were arranged as seen in the Figure 2.

The Cu plates were annealed in advance in an electric furnace type Carbolite CTF/12/75/700 at a temperature of 700 °C and kept for one hour, after which the copper plates were removed and cooled in water. The pieces were then placed one on top of the other, which then made a multilayer composite specimen (Figure 1). The samples were TIG-welded with additional aluminum (Figure 2) at both ends to minimize the risk of layer slippage.

Four cycles of annealing and hot rolling were performed (Figure 3). Before the hot-rolling cycle with a reduction of 50% by one pass, the composites samples were placed in the electric furnace for 1 h at 500 °C. The rolling multilayer composites samples were performed using a rolling machine with a rolling diameter of 220 mm and a rolling speed of 36 rpm.

Figure 4 shows the annealing of the copper layer at 700 °C and the 4 cycles of annealing at 500 °C followed by hot rolling. The elongation to failure increased after annealing.

After the first cycle of annealing and hot rolling, the copper layer underneath bonded. It can be deduced that the micro volumes at the interface did not adhere very well together, and the existence of some oxides can prevent bonding. Another hypothesis can be the different friction coefficient between the roll of the rolling mill and the copper layer and between the roll of the rolling mill and the aluminum layer.

The second cycle of annealing and hot rolling (with 3 layers) was carried out with a reduction of 50%. The sample with three layers was cut. The two pieces were stacked, and a sample with 6 layers was obtained, to achieve the third cycle of annealing and hot rolling with the same reduction percent.

In the fourth cycle of annealing and hot rolling, at the same reduction, the debonding of 2 layers (one of Cu and the other of Al) occurred. Therefore, a sample with 4 layers, Al-Cu-Al-Al, was obtained.

### 2.2. Material Characterisation

The sample prepared for SEM analysis was evaluated in the normal rolling direction, as were the EDX analyses performed to characterize the chemical composition and the elemental distribution in the Al-Cu multilayer composite, using the energy dispersive X-ray spectroscopy (EDX). For the SEM image recording, a JEOL JSM 5600LV electron microscope (Tokyo, Japan) was used, and for EDX, an Ulrim MAX 65 detector (Oxford Instruments, Aztech software, the version number 4.2) was used. A cross-sectional analysis was carried out to highlight the breakage of the specimen, on a Heckert 200 kN universal tensile-compression testing machine EDZ-20S (WEB Werkstoffprüfmaschinen, Leipzig, Germany) and an NI 6211 data acquisition board (National Instruments, Austin, TX, USA) for recording data.

The fracture surfaces after the tensile tests were also observed through SEM to determine the failure mode. Tensile testing, in accordance with the EN ISO 6892-1 test method at room temperature, was performed at 20 °C, and stress–strain curves were determined.

## 3. Results and Discussions

For the sample composites, after two cycles of annealing 1 h at 500 °C and hot rolling with 50% theoretical reduction, the dimensions obtained were 480 × 27 × 3 mm (Figure 5). Unfortunately, there was one Cu layer debonding and thus only three remaining layers: two layers of Al on the outside, and a layer of Cu in the center.

In order to test our hypothesis, a material with different characteristics and good adhesion between layers was needed to perform some destructive and non-destructive tests. The tensile test performed at 20 °C using a plate tensile specimen (Figure 6), with three layers, obtained after two cycles of annealing and hot rolling, highlighted a ductile fracture (Figure 7).

Figure 7 represents the room-temperature engineering stress–strain tensile test of the Cu-Al bimetallic specimen after the annealing and hot-rolling process. It can be seen that the first broken layer was the Al layer, from the outside, at a stress value of 157 MPa. The second breakage occurred simultaneously, within the two layers of Al and Cu, at 108 MPa.

In the fracture zone, the layers have unbonded. The elongation to failure, of the first aluminum layer, was increased after the first two cycles of annealing and hot rolling up to 23%. The elongation to failure of simultaneously broken Al-Cu layers was 34%.

It can be said that the mechanical properties obtained from engineering stress–strain curves are similar to those obtained in specialized literature by Taiqian Mo et al. [34] and Jun-peng Zhang et al. [35].

A comparison between the mechanical properties of the ultimate tensile strength of sample Al-Cu multilayered with ultimate theoretical tensile strength for each material Al (AlMgSi), and Cu-ETP is presented in Figure 8. As expected, the results of the tensile test of the Al-Cu multilayer composite specimen were closer to the tensile strength but still strongly influenced by the successive annealing process.

Figure 9 represents the sizes of the layers (Figure 9a) and the magnification of each layer, respectively (Figure 9b–d). It can be seen from the figures that each layer, after stretching, created an evident dimple. The area reduction of different layers (Figure 9) was calculated (from left to right Al 75.6%, Cu 81.9%, Al 80.03%), and the deformation of each layer was analyzed. After two cycles of annealing and hot rolling, the SEM analyses were conducted. From there, the deformation of the grains can be seen at the neck, indicated with white lines and dark dimples in all three layers of Al-Cu-Al. The fractography, in the cross section, is presented in Figure 9, and the elemental distribution map of Al-Cu multilayer composite (Figure 10) confirms the results of tensile test. The layers have separated in the fracture. SEM observations of the fracture behavior showed that the number of breaking germs (dimples) increased with the stretching of the material, which resulted in the ductile fracture of both materials.

The distribution maps presented in Figure 10 provides evidence of the adherence of the layers and the small diffusion of the interface.

The EDX line and area analyses shown in Figure 11 aim to characterize the Al-Cu multilayer composite from the point of view of chemical composition and to show the diffusion of the elements during the hot-rolling process. The area between the layer of the Al-Cu composite along the direction of rolling after two cycles was analyzed.

The analysis shows that at 50 μm (Figure 11) diffusion is present at the interface of Al-Cu and that the layers have a good adhesion.

In the multilayers made of the Al-Cu composite after two cycles of annealing–hot rolling, one of the Al layers debonded, necessitating the realization of new hot deformation cycles.

The third cycle was performed on a sample obtained by cutting and overlapping (see diagram of process Figure 3) the three previously rolling layers. Annealing for 1 h and severe rolling with a 50% theoretical reduction were also applied in this case.

The fourth cycle repeated the annealing–hot rolling in similar conditions. The layers size after the fourth cycle and the elemental map of the distribution of Al-Cu elements along the direction of rolling is presented in Figure 12. After the fourth cycle of annealing and hot rolling, the two Cu-Al layers below were debonded. The aluminum layers deformed differently from the copper layers. The Al-Cu TIG composite welding, which experienced problems with flatness during heating, can have been one of the causes of debonding. Through the distribution map of the Al-Cu composite and the SEM analysis, the remaining interfaces (four layers) connected were investigated.

Figure 12 illustrates that after four passes of the hot-rolling process, the Al-Cu multilayer composite has a heterogeneous structure, with discreet cracks at the interface bonding. The area reduction of different layers was calculated (from top to bottom Al 85.23%, Cu 86.86%, and Al 92.36%, and a second layer of Al 83.16%)), and the bonding interlayer was analyzed. The existence of microcracks at the interface creates the premise for the appearance of diffusion at the interface between the Al-Cu and Cu-Al layers. This phase, created at the interface between the layers, can be observed in the SEM micrographs recorded at higher magnification shown in Figure 13.

The chemical composition recorded in the key area of the interface is presented in Figure 14 alongside an image showing the phase contrast of the phases. Thus, in spectrum area 14 we have 99.2% Cu and 0.8% Al, in spectrum area 14 we have 81.4% Cu and 18.6% Al, in spectrum area 15 we have 92.6% Cu and 7.4% Al, in spectrum area 16 we have 53.6% Cu and 46.4% Al, and in spectrum 17 we have 7% Cu and 93% Al.

After the chemical composition analysis in the interface area, the possible phases (according to the Al-Cu binary phase diagram) appearing in each area spectrum numbering 13–17 can be seen in Table 5.

Binary phase diagrams are maps that represent the relationships between the temperature and the compositions and quantities of phases at equilibrium, which influences the microstructure of an alloy after William D. Callister, Jr. and David G. Rethwisch [44].

In Figure 15, the scan line drawn from the area of the Cu layer to the area of the Al layer (the same area as in the previous micrograph) also crosses the area with intermediate phases.

It can be concluded that the line scan confirms that over 36 μm diffusion occurred at the interface of the Al-Cu multilayer composite, with the formation of intermediate phases.

The microstructures and the EDX elemental mappings of the diffusion layers indicated that the metallurgical bonding interface was validated by the EDX presented in Figure 15. The bonding of the layers was closely related to the breaking stress, from the test to the tension. The role of diffusion in the fracture is to modify the properties of the interface layer (composite). With stronger bonding between the layers, we will be able to obtain increased resistance of the Al-Cu multilayer composite. However, it cannot be considered a failure even though the result shows that the outer layer, that of Al, gives way first. The further elongation of the remaining layers can lead us to think of applications in which the appearance of a crack or the breaking of a layer is of more concern, having the effect of raising an alarm signal before the final break.

## 4. Conclusions

Al-Cu sheets were fabricated via hot-rolling bonding at 500 °C with 50% theoretical rolling reductions.

The SEM and EDX images confirmed the ductile fracture predicted by the stress–strain curve. The specimen’s microstructures and mechanical properties were investigated. The findings are summarized below:The Al-Cu rolling sheets show poor interfacial bonding. A heterogeneous structure can be observed with discrete cracks and defects at the interface of the Al-Cu layers. With an increased number of cycles of the annealing–hot rolling of the multilayer Al-Cu composite, this becomes a more homogeneous structure.The rolling reduction has a significant effect on the deformation of Al-Cu hot-rolled sheets. The deformation of each layer was analyzed. Severe deformation by hot rolling leads to better bonding. As the number of severe hot rolling increases, diffusion occurs, and a better bond between them is obtained.The stress–strain curve indicates that the fracture appears first in the Al layer.The elongation to failure was increased after the first two cycles of annealing and hot rolling up to 23% (first break of aluminum) and was increased at 34% (second break).

## Figures and Tables

**Figure 1 materials-15-08807-f001:**
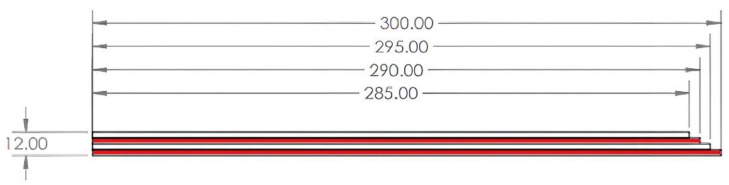
Initial sample of multilayer Al-Cu composite.

**Figure 2 materials-15-08807-f002:**
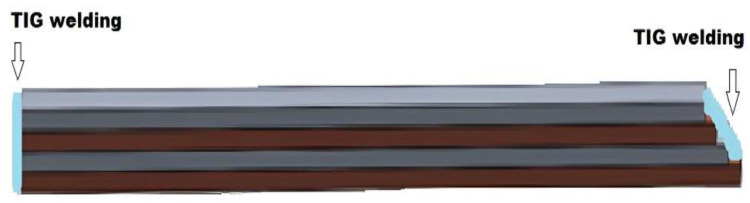
TIG welding with addition of aluminum at both ends of the initial sample.

**Figure 3 materials-15-08807-f003:**
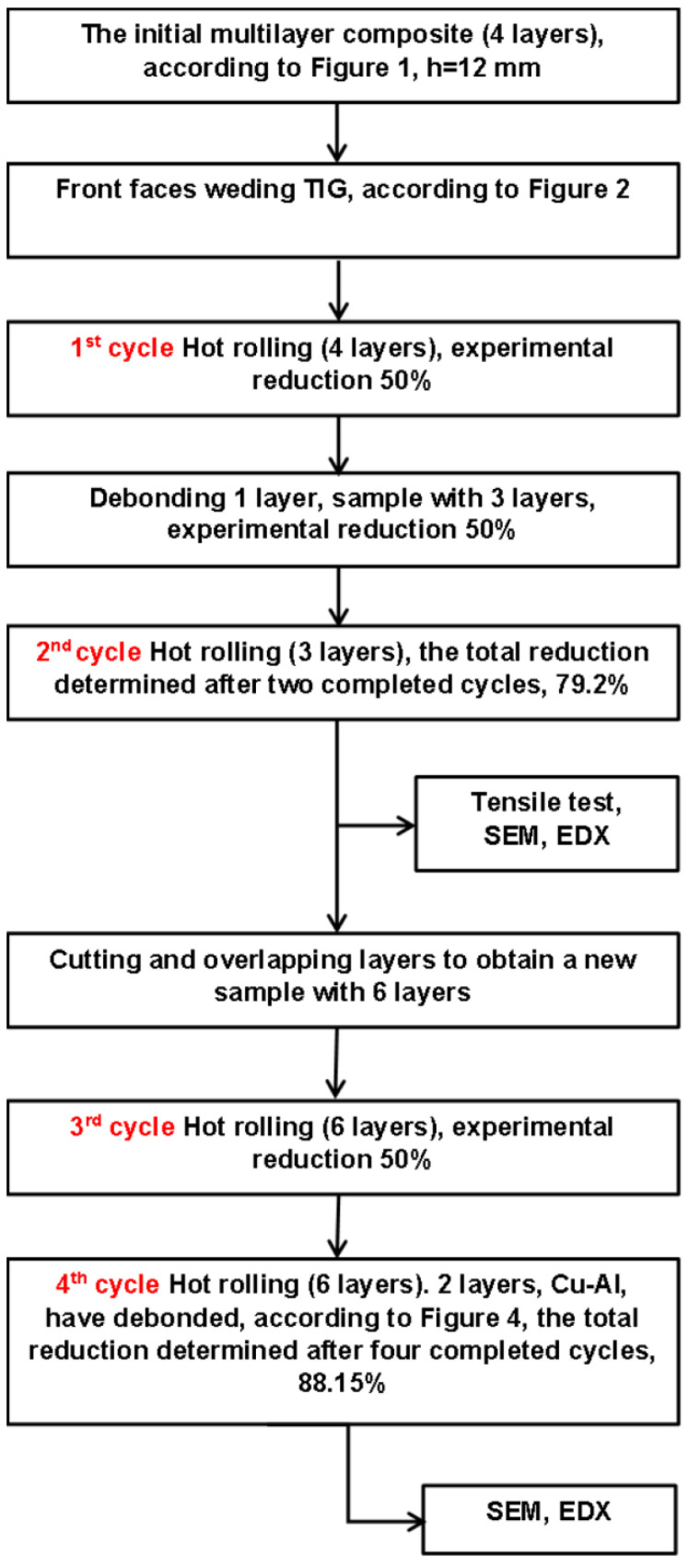
The process diagram.

**Figure 4 materials-15-08807-f004:**
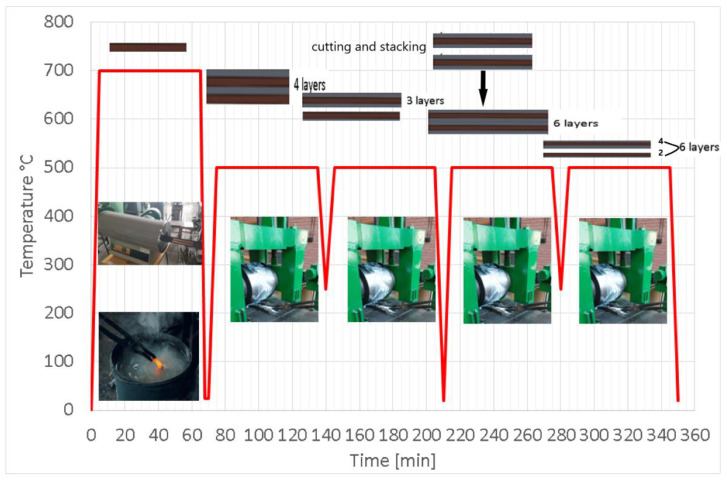
The temperature diagram.

**Figure 5 materials-15-08807-f005:**
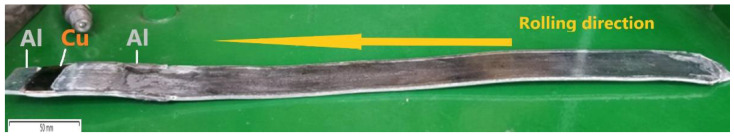
The sample composite after second cycle of rolling.

**Figure 6 materials-15-08807-f006:**
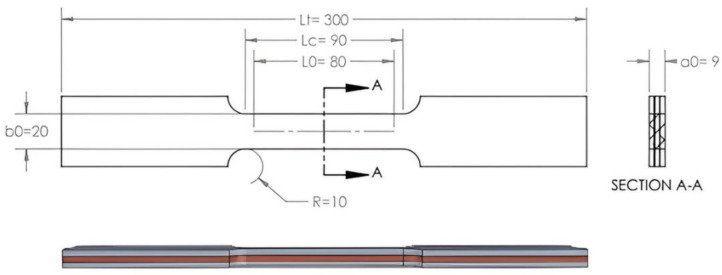
Plate tensile test specimen obtained after two cycles of annealing and hot rolling.

**Figure 7 materials-15-08807-f007:**
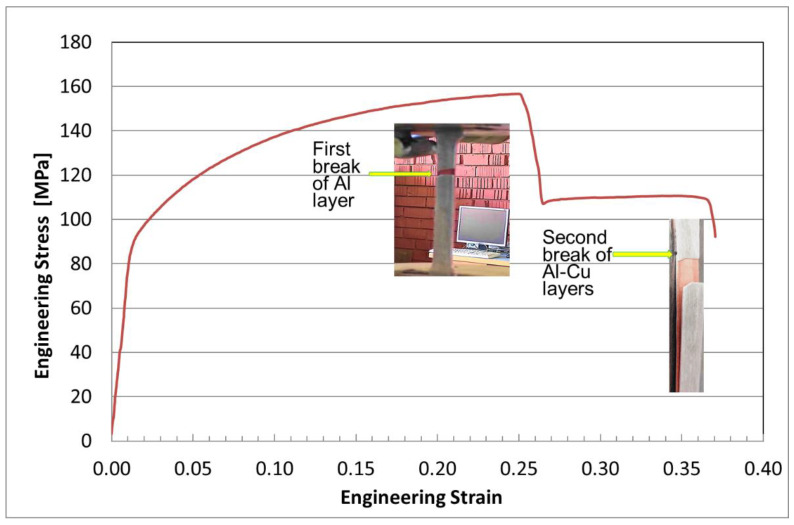
Stress–strain curve of Al-Cu processed through hot rolling.

**Figure 8 materials-15-08807-f008:**
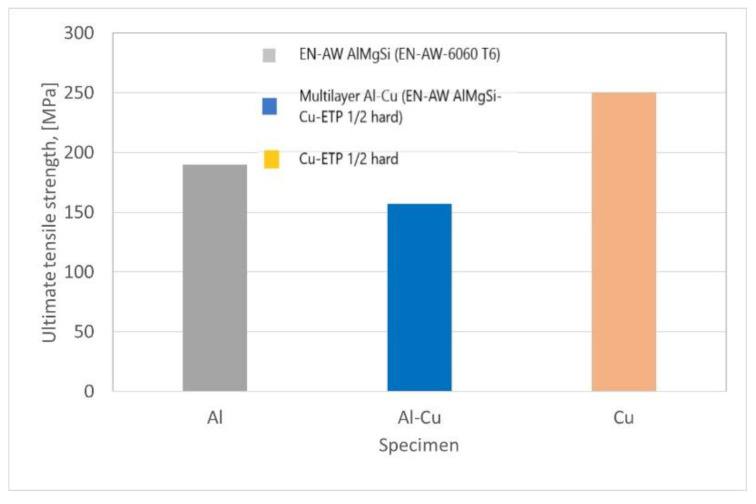
Comparison of the ultimate tensile strength and theoretical strength.

**Figure 9 materials-15-08807-f009:**
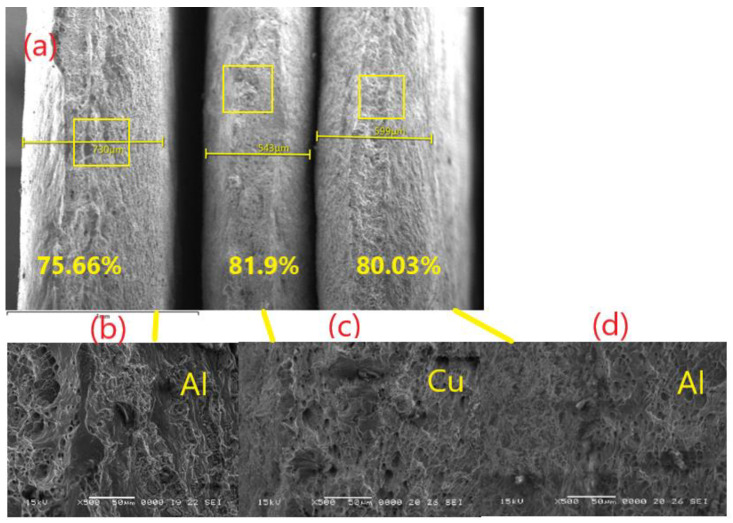
SEM micrographs of the tensile fracture surfaces after second cycle of anneling and hot rolling—low magnification ×35 with the percentage reduction calculated for each layer: (**a**) the multilayer composite, (**b**) EN-AW 6060 (AlMgSi)—high magnification detail—×500, (**c**) Cu-ETP ½ hard-high magnification detail—×500, and (**d**) EN-AW 6060 (AlMgSi) ×500.

**Figure 10 materials-15-08807-f010:**
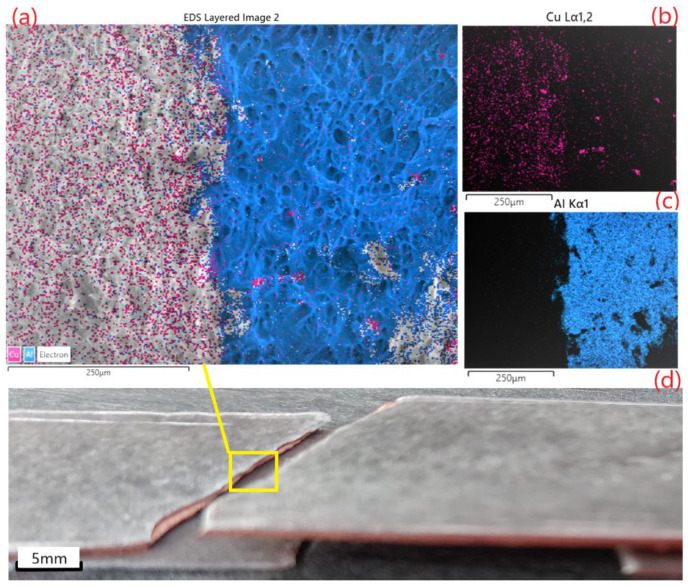
Elemental distribution map of Al-Cu composite after second cycle of anneling and hot rolling: (**a**) cross-section, (**b**) Cu, (**c**) Al, and (**d**) sample after tensile test.

**Figure 11 materials-15-08807-f011:**
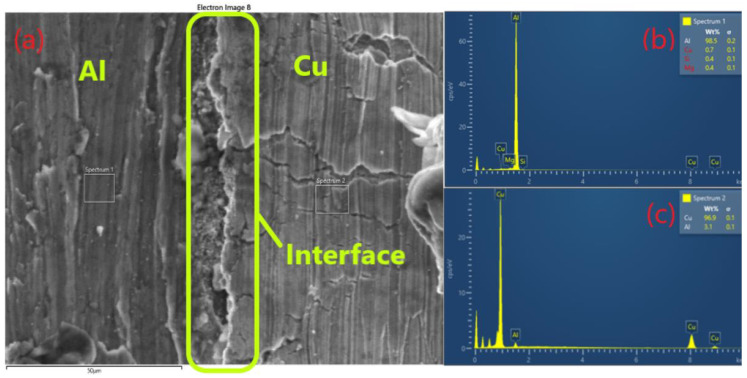
Bonding interface of the Al-Cu composite along the direction of rolling after two cycles (**a**); and EDX analysis of the two layers (**b**) and (**c**).

**Figure 12 materials-15-08807-f012:**
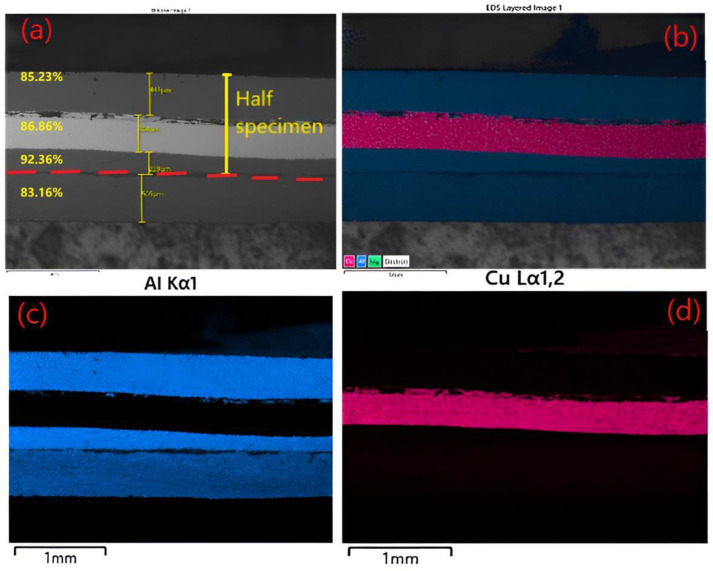
The layers size distribution (**a**) and (**b**) the elemental map of the distribution of Al-Cu elements along the direction of rolling after four cycles, (**c**) Al, and (**d**) Cu.

**Figure 13 materials-15-08807-f013:**
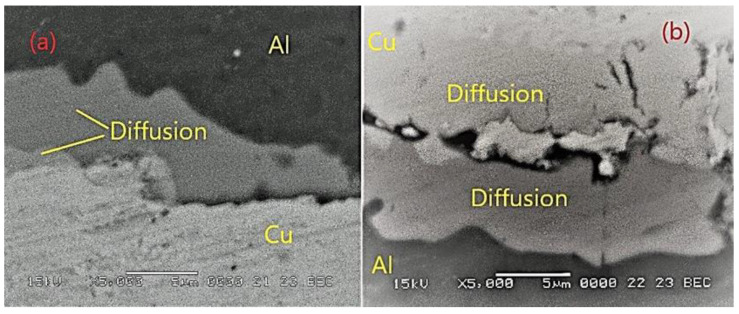
SEM image of Al-Cu composite after fourth cycle of anneling and hot rolling: (**a**) interface Al-Cu and (**b**) interface Cu-Al.

**Figure 14 materials-15-08807-f014:**
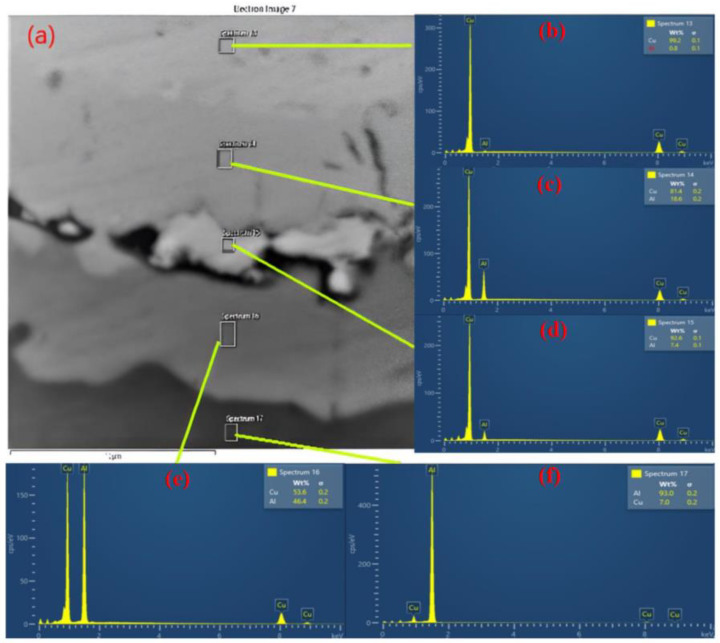
SEM micrographs with bounding interface of Al-Cu composite along normal rolling after 4 cycles (**a**) and EDX analisys at interface of layers: (**b**) spectrum area 13, (**c**) spectrum area 14, (**d**) spectrum area 15, (**e**) spectrum area 16, and (**f**) spectrum area 17.

**Figure 15 materials-15-08807-f015:**
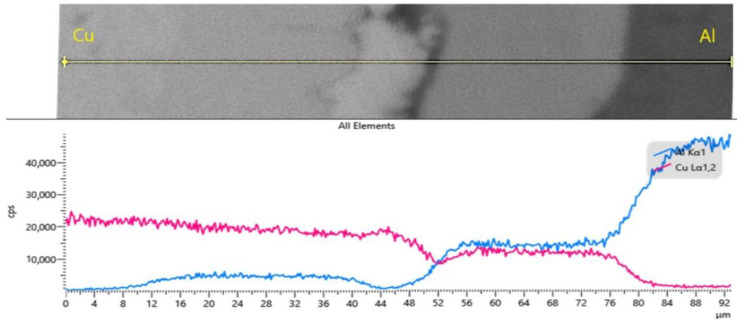
EDX line scanning across Cu-Al interfaces of rolling sheets.

**Table 1 materials-15-08807-t001:** The chemical composition (wt.%) EN-AW-6060 (EN-AW AlMgSi) according to EN573-3 [40].

Si	Fe	Cu	Mn	Mg	Cr	Zn	Ti	Other	Al
0.3–0.6	0.1–0.3	0.10	0.10	0.35–0.6	0.05	0.15	0.1	0.05	Remainder

**Table 2 materials-15-08807-t002:** Chemical composition Cu-ETP ½ hard according to EN 1652 [41] (wt.%).

Cu	O	Si	Bi	Pb	Other Elements
99.90	0.040	maximum 0.015	0.0005	0.005	Remainder

**Table 3 materials-15-08807-t003:** Mechanical properties EN-AW-6060 (EN-AW AlMgSi) according to EN755-2 [42].

Temper A50 (%)	Wall Thickness, e (mm)	Yield Stress Rp_0.2_ (MPa)	Tensile Strength, Rm (MPa)	Elongation		Hardness (HB)
				A (%)	A50 (%)	
T6	<5	150	190	8	6	65

**Table 4 materials-15-08807-t004:** Mechanical properties Cu-ETP ½ hard according to EN 13601 [43].

Material	Tensile Strength, Rm (MPa)	0.2% Proof Strength Rp_0.2_ (MPa)	Elongation		Hardness	
			A_100 mm_ (%)	A (%)	Brinell HBW	Vickers HV
R250 ½ Hard	250	minimum 180	8	12	65–90	70–95

**Table 5 materials-15-08807-t005:** The possible phases (according to the Al-Cu binary phase diagram).

Area Spectrum No.	Al (wt.%)	Cu (wt.%)	Possible Phases
13	99.2	0.8	Cu(α) + Al(α)
14	81.4	18.6	Cu + Al_2_Cu + Al_5_Cu_8_
15	92.6	7.4	Cu + Al_2_Cu + Al_5_Cu_8_
16	53.6	46.4	Al_2_Cu + Al_9_Cu_11_
17	93	7	Cu + Al_2_Cu

## Data Availability

Not applicable.

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
