# Peer review of "Studies on Hot-Rolling Bonding of the Al-Cu Bimetallic Composite"

_materials, 2022, doi:10.3390/ma15248807_

Round 1
Reviewer 1 Report
This manuscript presents the investigation on the bonding of Al-Cu bimetallic composite layers after hot rolling and the behaviour of the diffusion at the interface. However, the novelties of this research work are unclear which need to be further clarified. The discussion is too plain to be considered for a journal article, and the conclusions are too common for the materials engineering research field. In addition, the language of the description throughout the manuscript is poor together with many errors and grammar issues. Many words and description in some figures of this manuscript are present badly. Regarding these, I have to make a decision that it is not recommended to be published in this journal.
Author Response
Dear Editors and Reviewer 1,
I would like to thank you for your suggestions and for your significant contribution in improving this paper. We are deeply grateful to you for all your suggestions and help, and hope that the correction will meet with your approval.
All changes are highlighted in red in the paper.
Please allow us to respond point-by-point to your observations:
Comments and Suggestions for Authors
This manuscript presents the investigation on the bonding of Al-Cu bimetallic composite layers after hot rolling and the behaviour of the diffusion at the interface. However, the novelties of this research work are unclear which need to be further clarified. The discussion is too plain to be considered for a journal article, and the conclusions are too common for the materials engineering research field. In addition, the language of the description throughout the manuscript is poor together with many errors and grammar issues. Many words and description in some figures of this manuscript are present badly. Regarding these, I have to make a decision that it is not recommended to be published in this journal.
From the authors' point of view, the work has undergone "major revision". We would especially appreciate it if you would consider the changes made in the manuscript.
The novelties of this research work were:
This study gives a description of the fabrication of Al-Cu composite (Aluminum alloy 6060 plate (EN-AW AlMgSi) and Cu-ETP ½ hard (CW004A)), by hot rolling. The samples were made of four layers, 4 cycles of annealing and hot rolling with severe reduction (50%).
An attempt was made to analyze the bonding of Al-Cu bimetallic composite layers and the highlight of the diffusion at the boundary between the layers.
Following this, the elemental diffusion between layers is discussed while the fracture behavior is also analyzed.
The Al-Cu composite material obtained was analyzed by scanning electronic microscopy (SEM) analysis, after being subjected to the tensile test, as well as Energy Dispersive X-Ray (EDX) analysis.
The deformation of each layer was analyzed.
The discussion section has been improved.
It was tried to clarify the Cu layer detachment from the second cycle as well as the Cu-Al layers detachment from the fourth cycle.
The conclusions have been modified.
The language improvement was made according to the authors.
Thank you for your pertinent proofreading suggestions
Your comments are of great significance to our manuscript. Thank you for your careful review. We sincerely hope that you can accept our response.

Reviewer 2 Report
This study gives a description of the fabrication of Al-Cu composite. An attempt was made to analyze the bonding of Al-Cu bimetallic composite layers and the highlight of the diffusion at the boundary between the layers, by hot rolling. Aluminum alloy 6060 plate (EN-AW AlMgSi) and Cu-ETP ½ hard (CW004A) 16 plate were used. Following this, the elemental diffusion between layers is discussed while the fracture behavior is also analyzed. However, some comments should be addressed before publication.
1. In the introduction, the background and research progress of Al-Cu composite should be given.
2. Line 190-196, how to determine the type of phases should be explained.
3. In Figure 8, the area reduction of different layers can be calculated and, in turn, the deformation of each layer can be analyzed.
4. There is diffusion between layers. What is the role of this diffusion in the fracture?
5. If possible, the tensile curve of the six layers should be given.
Author Response
Dear Editors and Reviewer 2,
I would like to thank you for your suggestions and for your significant contribution in improving this paper. We are deeply grateful to you for all your suggestions and help, and hope that the correction will meet with your approval.
All changes are highlighted in red in the paper.
Please allow us to respond point-by-point to your observations:
Comments and Suggestions for Authors
This study gives a description of the fabrication of Al-Cu composite. An attempt was made to analyze the bonding of Al-Cu bimetallic composite layers and the highlight of the diffusion at the boundary between the layers, by hot rolling. Aluminum alloy 6060 plate (EN-AW AlMgSi) and Cu-ETP ½ hard (CW004A) 16 plate were used. Following this, the elemental diffusion between layers is discussed while the fracture behavior is also analyzed. However, some comments should be addressed before publication.
All changes were made with "Track changes".
The language has been improved throughout the manuscript.
1. In the introduction, the background and research progress of Al-Cu composite should be given.
A paragraph with the background of the works in the field of Al-Cu composites was introduced (lines 50-58).
2. Line 190-196, how to determine the type of phases should be explained.
A new Table 5 has been inserted for a better representation of the possible phases determined from the Al-Cu binary phase diagram, line 241-247.
3. In Figure 8, the area reduction of different layers can be calculated and, in turn, the deformation of each layer can be analyzed.
Figure 8 became Figure 9 with the modifications of the calculated area reduction on each layer. The same area reduction calculation was made for figure 12.
4. There is diffusion between layers. What is the role of this diffusion in the fracture?
Yes, it is diffusion between layers. The role of diffusion in the fracture is to modify the properties of the interface layer (composite).
5. If possible, the tensile curve of the six layers should be given.
Unfortunately, the sample obtained after the fourth annealing and hot rolling cycle does not have the necessary length to obtain a tensile sample.
Thank you for your pertinent proofreading suggestions
Your comments are of great significance to our manuscript. Thank you for your careful review. We sincerely hope that you can accept our response.

Reviewer 3 Report
The presented manuscript seems to be interesting for readers of the Materials journal, it is written in a good manner and suits the requirements of the journal. It can be accepted for publication after minor corrections listed below.
- English language of manuscript is acceptable in general. However, it would be much better to improve. Please avoid the unnecessary long sentence. Also, some grammatical and typos mistakes can be observed. For example: plat band ، Accrossing، Traction test؛ fault surfaces.
- Line 68: H202 should be replaced by H2O2 or H2O
- Tables 1 and 3 should be given "percentage by weight" in the superscript or text of the table
- In the flowchart of figure 3, in the row one before the last, reference should be made to figure 4
- Figure 5 should be scaled
- The captions of all figures (especially figures 5 and 6) should be corrected and completed and the details of the sample and different parts of the figure should be explained.
- The order of reference to figures should be based on their placement in the text, for example figure 4 is referenced before figure 3 and should be corrected.
-The cause of Cu layer debonding should be investigated with complete characterization and the efforts made to solve this problem should be mentioned in the text of the article.
- Microscopic images show the complete separation of the layers from the interface; Contrary to the claim shown in Figures 9 and 10, it is suggested that similar to the microstructure shown in Figure 11, before performing the tensile test in the second cycle, the macro and microstructure of the joint should be examined.
- In Figure 11, the number and repetition of the layer is different from the content presented in Section 2. The copper layer should be one in the middle of the aluminum layer, while in the figure only one copper layer is observed.
- In line 159, the following sentence is unintelligible and should be rewritten:
- The analysis shows that at about 50 μm the diffusion is not present in the layered structure and the layers have a good adhesion.
- It is suggested to present the numerical results of EDX analysis at the interface of layers and the possible phases related to it in a table attached to Figure 13.
- To prove the claim of the authors regarding the improvement of tensile properties after 4 rolling cycles, it is necessary to provide the stress-strain diagram and tensile test results after each rolling stage (compared to Figure 7).
- The literature review is not thorough enough. It is crucial to stress the novelty of this study. Therefore I strongly recommend expanding the literature review. The paper has some new original investigation and deserves to emphasize its originality. For a more detailed review of the literature, I recommend referring to the following papers;
[]Metals, 10, 2020, 634
[]Journal of Manufacturing Processes 70 (2021): 152-162.
[]Advanced Processes in Materials Engineering 10.2 (2016): 131-142.
.
Author Response
Dear Editors and Reviewer 3,
I would like to thank you for your suggestions and for your significant contribution in improving this paper. We are deeply grateful to you for all your suggestions and help, and hope that the correction will meet with your approval.
All changes are highlighted in red in the paper.
Please allow us to respond point-by-point to your observations:
Comments and Suggestions for Authors
The presented manuscript seems to be interesting for readers of the Materials journal, it is written in a good manner and suits the requirements of the journal. It can be accepted for publication after minor corrections listed below.
- English language of manuscript is acceptable in general. However, it would be much better to improve. Please avoid the unnecessary long sentence. Also, some grammatical and typos mistakes can be observed. For example: plat band ،Accrossing ،Traction test؛ fault surfaces.
All changes were made with "Track changes".
The language has been improved throughout the manuscript.
A paragraph with the background of the works in the field of Al-Cu composites was introduced (lines 50-58).
- Line 68: H202 should be replaced by H2O2 or H2O
The H2O2 formula was replaced in line 76.
- Tables 1 and 3 should be given "percentage by weight" in the superscript or text of the table
"(wt.%) " was added in the superscript of Tables 1 and 3, line 86 and 92.
- In the flowchart of figure 3, in the row one before the last, reference should be made to figure 4
Figure 3 has been modified referring to Figure 4, and additions regarding the process diagram (line 110).
Figure 4 has been modified. The figure illustrates more clearly the stacking of layers and the effort to achieve bonding.
- Figure 5 should be scaled
Figure 5 has been scaled.
- The captions of all figures (especially figures 5 and 6) should be corrected and completed and the details of the sample and different parts of the figure should be explained.
Figure 5 has been completed and the details of the sample have been explained (line 142-145).
- The order of reference to figures should be based on their placement in the text, for example figure 4 is referenced before figure 3 and should be corrected.
The references were placed in the text in the order of appearance, as close as possible to the related figure.
-The cause of Cu layer debonding should be investigated with complete characterization and the efforts made to solve this problem should be mentioned in the text of the article.
It was tried to clarify the Cu layer detachment from the second cycle as well as the Cu-Al layers detachment from the fourth cycle, in the following rows: lines 112-115 and lines 117-121, also in 212-219.
- Microscopic images show the complete separation of the layers from the interface; Contrary to the claim shown in Figures 9 and 10, it is suggested that similar to the microstructure shown in Figure 11, before performing the tensile test in the second cycle, the macro and microstructure of the joint should be examined.
A new diagram (Figure 8) comparing the ultimate tensile strength and theoretical strength has been introduced.
Figures have been renumbered, Figures 9 -Figure 15.
Clarifications in line 168-196.
- In Figure 11, the number and repetition of the layer is different from the content presented in Section 2. The copper layer should be one in the middle of the aluminum layer, while in the figure only one copper layer is observed.
Clarifications in line 212-217.
- In line 159, the following sentence is unintelligible and should be rewritten:
Debonded, unbonded.
- The analysis shows that at about 50 μm the diffusion is not present in the layered structure and the layers have a good adhesion.
The phrase has been reformulated, line 204-205
- It is suggested to present the numerical results of EDX analysis at the interface of layers and the possible phases related to it in a table attached to Figure 13.
A new Table 5 has been inserted for a better representation of the possible phases determined from the Al-Cu binary phase diagram.
- To prove the claim of the authors regarding the improvement of tensile properties after 4 rolling cycles, it is necessary to provide the stress-strain diagram and tensile test results after each rolling stage (compared to Figure 7).
A new diagram (figure 8) comparing the ultimate tensile strength and theoretical strength has been introduced.
- The literature review is not thorough enough. It is crucial to stress the novelty of this study. Therefore I strongly recommend expanding the literature review. The paper has some new original investigation and deserves to emphasize its originality. For a more detailed review of the literature, I recommend referring to the following papers;
[]Metals, 10, 2020, 634
[]Journal of Manufacturing Processes 70 (2021): 152-162.
[]Advanced Processes in Materials Engineering 10.2 (2016): 131-142.
The list of references from 26-39 and 44 has been enriched. The citations are found in the main text in the introduction section.
Thank you for your pertinent proofreading suggestions
Your comments are of great significance to our manuscript. Thank you for your careful review. We sincerely hope that you can accept our response.

Round 2
Reviewer 1 Report
Looks better.
Reviewer 2 Report
The comments are all well addressed.
Reviewer 3 Report
The authors have addresses the most of the corrections required by the Reviewers. So, it can be accepted for publication now.